# Early Expression of Tet1 and Tet2 in Mouse Zygotes Altered DNA Methylation Status and Affected Embryonic Development

**DOI:** 10.3390/ijms23158495

**Published:** 2022-07-31

**Authors:** Qi Qi, Qianqian Wang, Kailing Liu, Jiangyue Bian, Zhixuan Yu, Jian Hou

**Affiliations:** State Key Laboratory of Agrobiotechnology and College of Biological Science, China Agricultural University, Beijing 100193, China; qiqi2017@cau.edu.cn (Q.Q.); wangqianqianhp@163.com (Q.W.); goodluck-lkl@hotmail.com (K.L.); jiangyuebian@cau.edu.cn (J.B.); ayumica@163.com (Z.Y.)

**Keywords:** DNA methylation, mice, zygotes, Tet1, Tet2, embryonic development

## Abstract

Ten-eleven translocation (Tet) dioxygenases can induce DNA demethylation by catalyzing 5-methylcytosine(5mC) to 5-hydroxymethylcytosine(5hmC), and play important roles during mammalian development. In mouse, Tet1 and Tet2 are not expressed in pronucleus-staged embryos and are not involved in the genomic demethylation of early zygotes. Here, we investigated the influence of Tet1 and Tet2 on methylation of parental genomes by ectopically expressing Tet1 and Tet2 in zygotes. Immunofluorescence staining showed a marked 5hmC increase in the maternal pronucleus after injection of Tet1 or Tet2 mRNA into zygotes. Whole-genome bisulfite sequencing further revealed that Tet2 greatly enhanced the global demethylation of both parental genomes, while Tet1 only promoted the paternal demethylation. Tet1 and Tet2 overexpression altered the DNA methylation across genomes, including various genic elements and germline-specific differently methylated regions. Tet2 exhibited overall stronger demethylation activity than Tet1. Either Tet1 or Tet2 overexpression impaired preimplantation embryonic development. These results demonstrated that early expression of Tet1 and Tet2 could substantially alter the zygotic methylation landscape and damage embryonic development. These findings provide new insights into understanding the function of Tet dioxygenases and the mechanism of DNA methylation in relation to embryogenesis.

## 1. Introduction

DNA methylation at the five position of cytosine (5mC) is an important epigenetic modification that regulates chromatin structure and gene expression. In mammals, DNA methylation plays a crucial role in gene imprinting, X-chromosome inactivation, genomic stability, and embryonic development [1,2]. DNA methylation is maintained at higher levels in mature gametes and differentiated somatic cells, but it undergoes genome-wide reprogramming in preimplantation embryos and primordial germ cells (PGCs) [1,2,3]. In the mouse zygotes, the paternal genome is actively demethylated shortly after fertilization, while demethylation of the maternal genome takes place gradually in a replication-dependent manner [4,5]. The global DNA demethylation of parental genomes is essential in order for embryos to acquire developmental totipotency [1].

It has been demonstrated that active DNA demethylation can be achieved by the conversion of 5mC to 5-hydroxymethylcytosine(5hmC), which is mediated by ten-eleven translocation (Tet) dioxygenases [6,7]. Tet proteins can catalyze iterative oxidation of 5mC to 5hmC, 5-formylcytosine (5fC), and 5-carboxylcytosine (5caC) [8,9]. The oxidized derivatives (mainly 5fC and 5caC) can be excised by thymine-DNA glycosylase (TDG), or they are passively diluted by DNA replication during cell division, both of which result in the removal of methylation and restoration of unmodified cytosines [9,10,11]. These pathways have been explored to explain the active demethylation mechanism in mammalian embryos [12,13].

The Tet dioxygenase family consists of Tet1, Tet2, and Tet3, but only Tet3 is specifically expressed in oocytes and is responsible for the active demethylation of the paternal genome in mouse zygotes [14,15,16]. Tet3 is predominantly located in the paternal pronucleus and exerts a demethylation function [12,16,17]. This paternal-specific recruitment of Tet3 was suggested to be attributed to Stella (Dppa3/PGC7) stored in oocytes and zygotes; Stella can tightly bind to dimethylated histone 3 lysine 9 (H3K9me2) enriched in the maternal chromatin and inhibits the recruitment of Tet3 to the maternal pronucleus, preventing maternal DNA demethylation [18].

In contrast with Tet3, Tet1 and Tet2 are highly expressed in cleavage-stage embryos, but not in oocytes [14,19], indicating that they probably do not participate in demethylation in the pronucleus. Moreover, Tet1 or Tet2 seem to be dispensable for embryonic development, as deletion of Tet1 or Tet2 (alone or both) does not affect the full-term development of embryos to birth, and the mutant mice were even viable [19,20,21,22]. Both Tet1 and Tet2 are abundantly present in PGCs and are required for the demethylation of imprints and germline-specific genes [23,24,25,26]. However, Tet1 and Tet2 do not seem to fully compensate for Tet3, because knockout of the maternal-Tet3 alone causes perinatal lethality of pups [16,19]. Thus, Tet protein members appear to have distinct roles in genetic demethylation and biological processes, but their action needs further investigation.

Several questions need to be addressed regarding the function of Tet1 and Tet2 in early embryos. First, as Tet1 and Tet2 are not present in oocytes and pronucleus-stage embryos, it is unknown to what extent these proteins influence the demethylation of zygotic genomes if they are forced to express in early embryos. Second, given that Stella is recruited specifically in the maternal pronucleus and inhibits Tet3′s activity [18], can it suppress the function of Tet1 and Tet2 in demethylating the maternal genome as it does for Tet3? Third, if Tet1 and Tet2 can induce methylation changes at the pronucleus stage, do these changes influence embryo development? To answer these questions, in this study, we overexpressed Tet1 or Tet2 in mouse pronucleus-stage embryos and evaluated their effects on genomic methylation and embryo development. Our data provide new information for understanding the functions of Tet proteins in mammalian embryos.

## 2. Results

### 2.1. Overexpression of Tet1 and Tet2 in Zygotes Increased the DNA Oxidation of the Maternal Genome

We first confirmed the expression profile of Tet family members in mouse preim-plantation embryos and found that only Tet3 was highly expressed in oocytes and zygotes, whereas Tet1 and Tet2 were expressed at the cleavage stages (Appendix A). This result was in line with previous studies [14,16,19].

To test the action of Tet1 and Tet2 in the pronucleus-stage embryos, their mRNAs were microinjected into mouse zygotes, and GFP mRNA was used as an injection control. The embryos were immunostained 12 h after injection of the mRNA. Asymmetric methylation patterns between the two parental pronuclei were observed in either the non-injection control or the GFP injection group, with high 5mC levels in the female pronucleus but high 5hmC levels in the male pronucleus (Figure 1A), which is consistent with previous reports [14,15,16]. No difference in 5hmC levels was observed between the GFP group and non-injection control (Figure 1A), confirming that GFP injection did not influence the zygotic demethylation. In the Tet1 and Tet2 injection groups, the global 5hmC levels were markedly increased in the female pronucleus, which was quite different from that in the GFP group (Figure 1A), suggesting that Tet1 and Tet2 have an oxidation function on the maternal genome. We found that overexpression of Tet1 and Tet2 did not affect the overall enrichment and distribution of Stella and H3K9me2 in pronuclei (Figure 1B). However, we did not find apparent changes in 5mC levels after Tet mRNA injection.

### 2.2. Whole-Genome Bisulfite Sequencing (WGBS) Analysis of the Methylome

We used WGBS analysis of the methylome to precisely investigate the DNA methylation at the base-resolution level. To distinguish between the parental genomes, two mouse strains with different single-nucleotide polymorphisms (SNPs), DBA/2J (♂) and C57BL/6J (♀), were mated to produce hybrid embryos, and the paternal and maternal genomes in the hybrid embryos were distinguished by sequence alignment (n = 2,917,492). It should be noted that bisulfite treatment cannot distinguish between 5mC and 5hmC [27]. Hence, the methylation data used in the subsequent sections reflect the 5mC + 5hmC level among the CG sequence environment.

Approximately 65 million mapped reads with 288 million C-sites (≥5×) in five sequenced samples were obtained from the raw data. Across the three zygote samples, we obtained a total of 6,798,610 mapped reads with 30,954,235 C-sites (≥3×) based on the call-SNPs (Appendix A).

Having confirmed the successful generation of the WGBS datasets for all samples, we first focused our analyses on the mean DNA methylation. The sperm methylation level (75.78%) was much higher than the oocyte methylation level (40.32%). After fertilization, the paternal genome was considerably demethylated (the methylation level decreased to 50.64%), while the methylation level of the maternal genome showed little change (slightly increased to 43.58%) (Figure 2A and Appendix A), which is in agreement with previous studies showing that global demethylation occurs specifically in the paternal genome [28,29].

Relative to the GFP injection, injection of Tet1 or Tet2 resulted in a 3.41% or 22.79% zygotic methylation decrease, respectively. Interestingly, in the Tet1-injected zygotes, the methylation levels were downregulated by 16.71% in the paternal genome but upregulated by 4.98% in the maternal genome compared to the GFP-injected embryos (Figure 2A). However, the methylation levels of paternal and maternal genomes in the Tet2-injected zygotes were reduced by 35.54% and 12.15%, respectively, and the proportions of hypomethylated sites were significantly increased in comparison to the GFP injection control (Figure 2A). Exploration of a representative region (chr19: 36,372671–49,248935) reflected the same situation as the mean methylation (Figure 2B). Thus, Tet1 and Tet2 can induce demethylation on the paternal genome, but Tet2 has stronger demethylation activity than Tet1. Tet2 can also demethylate the maternal genome, albeit to a lesser extent than on the paternal genome, but Tet1 slightly increased the maternal methylation.

### 2.3. Effects of Tet1 and Tet2 on Various Genomic Regions in Zygotes

We mapped Tet1- and Tet2- induced DNA demethylation to different genetic elements. For the paternal genome during the transition from sperm to zygote pronucleus, almost all genetic regions, including functional regions, intergenic regions, and transposon regions, showed loss of DNA methylation in all three groups of zygotes (Figure 2C), consistent with the trend of average methylation levels (Figure 2A). Compared to GFP injection, both Tet1 and Tet2 injection further downregulated the methylation levels of the paternal genome in all regions, and Tet2 yielded more potent demethylation than Tet1 (Figure 2C). The demethylation of repetitive sequences was also enhanced by overexpression of Tet1 and Tet2, particularly in the LTR regions, which includes elements of endogenous retroviral locus (ERVs) and intracisternal A particles (IAPs) (Figure 2D).

For the maternal genome, the GFP injection group showed no significant changes in DNA methylation levels in most functional regions compared to oocytes, but a slight methylation increase occurred in the promoter and intergenic regions, including repetitive sequences (Figure 2B). The maternal genome in the Tet1 group generally showed similar methylation patterns to that of the GFP group, but had a higher methylation level in intergenic regions, repetitive sequences, and promoter regions (Figure 2C). In contrast, the Tet2 group exhibited much more demethylation in all regions in the maternal genome than the GFP and Tet1 groups, including the subcategories of LTRs (Figure 2C,E).

### 2.4. Methylation Sites Affected by Tet1 and Tet2

To further determine the roles of Tet1 and Tet2 on the parental genome, the methylation-alerted sites in different genomic regions were counted. Differentially methylated sites were defined as a more than 20% methylation change in differentially methylated regions (DMRs), with those higher than the GFP group considered hypermethylated sites and, conversely, hypomethylated sites.

In the paternal genome, analysis of methylation changes in genic regions revealed that Tet1 overexpression resulted in significant hypomethylation of 1381 sites and hypermethylation of 49 sites, while Tet2 overexpression led to hypomethylation of 5725 sites (Figure 3A and Appendix A). The two Tet groups shared 487 identical hypomethylation sites, accounting for 35.26% of the total hypomethylation sites in genic regions in the Tet1 group (Figure 3A and Appendix A). In repeat sequences, 832 sites were hypomethylated in the Tet1 group, and 4867 sites were hypomethylated in the Tet2 group, of which 210 sites overlapped between the two groups (Figure 3B and Appendix A), accounting for 25.24% of the hypomethylated sites in the repeats in the Tet1 group.

In the maternal genome, Tet1 overexpression resulted in more hypermethylated sites than hypomethylated sites (1715 vs. 675), but Tet2 overexpression led to the overwhelming majority of hypomethylated sites to hypermethylated sites (3980 vs. 361) (Figure 3A). The two groups shared 111 hypomethylated sites and 33 hypermethylated sites. The average overlapping ratio of hypomethylated sites was much lower in the maternal genome (111/675, 16.44%) than in the paternal genome (35.26%) (Figure 3A and Appendix A). A similar situation was also observed for the repeats in the maternal genome (Figure 3B and Appendix A). Together, the results of paternal and maternal genomes suggest that Tet1 and Tet2 had different action biases on genomic loci, although they also targeted some common sites.

### 2.5. The Propensity of Tet1 and Tet2 to Base Sequences

To gain a comprehensive understanding of whether Tet1 and Tet2 have a propensity to recognize and interact with DNA sequences, we compared the CG dinucleotide-embedded sequences, i.e., NNCGNN, where the CG methylation was altered by the action of Tet proteins, and analyzed the frequency of individual bases in 10 bp sequences flanking CG. The observed/expected (obs/exp) value of the four types of bases for each site was the probability of the site occurring at a given position. Its variance indicated the propensity for the sites of action.

From the results, we found that the NNCGNN sequences, where the parental genome hypomethylation occurred, showed a tendency for Tet1 and Tet2 to have symmetrical distributions at positions −5 to +5 (Figure 3C). For the position −5 or +5 in the paternal genome, the variance of Tet1 was greater than that of Tet2 (Figure 3C). Notably, Tet2 showed a stronger tendency than Tet1 at the −1 and +1 positions in the maternal genome, but Tet1 had a stronger role than Tet2 at the more distant position (−5) in both paternal and maternal genomes (Figure 3C,D). However, for the hypermethylated CG sequences, both parental genomes had a higher preference for Tet1 at position +5 and a preference for Tet2 at position −5 (Figure 3E,F). In contrast with the sequences of hypomethylated CG, both parental genomes had a relatively lower preference for the nearest position of the hypermethylated CG (Figure 3C–F).

We then examined the propensity for individual bases from the −5 to +5 positions of the CG. In both parental genomes, the distribution of bases in either Tet1- or Tet2- acting DNA sequences were characterized by palindromes, e.g., a more significant proportion of base A and C at −1 and the highest levels of G and T at +1, as shown in Figure 3I,J for the hypermethylated sites. Similar patterns were also found in the sequences of hypomethylated CG sites in both parental genomes (Figure 3G,H). Tet proteins preferred A or G for bases at −1, C at −2 and G at −5, with the opposite position being the complementary base to these clips (Figure 3G,H). This suggested that the sequences acted by Tet1 and Tet2 tended to be those with higher CG content.

### 2.6. Influences of Tet1 and Tet2 on Germline-Specific DMRs

Previous studies have identified many DMRs between oocytes and sperm, which were either hypermethylated (methylation levels > 75%) or hypomethylated (methylation levels < 25%) in both gametes, named germline-specific DMRs (gDMRs) [29,30,31].

Using SNPs, we identified 16,583 sperm-specific DMRs (sDMRs) and 100 oocyte-specific DMRs (oDMRs), and tracked their methylation status in zygotes (Figure 4A,B). We found that the methylation levels of sDMRs were markedly decreased in the paternal pronucleus, and Tet1 or Tet2 injection resulted in a greater decrease than GFP injection (Figure 4A). However, the sDMR methylation levels were slightly but significantly increased in the maternal pronucleus in all three groups, with the most significant increase in the Tet1 group (Figure 4C). The methylation of oDMRs was not dramatically changed in the maternal and paternal genomes in GFP- and Tet1-injected zygotes, but it was downregulated considerably in Tet2-injected zygotes (Figure 4B,D). These results suggest that the sDMRs underwent a dramatic methylation change during the gamete to zygote transition, which involved paternal demethylation and maternal remethylation. In contrast, the oDMRs showed minor changes after fertilization. In either case, Tet2 exhibited more potent demethylation than Tet1.

### 2.7. Roles of Tet1 and Tet2 on Imprinting Control Regions (ICRs)

Gene imprinting is critical for embryonic development. Some DMRs are classified as germline imprinting control regions (gICRs) and are resistant to demethylation during early embryogenesis [30,31]. We examined the methylation of 25 maternally imprinted gICRs and 3 paternally imprinted gICRs identified previously in mice [31], and confirmed their differential methylation between oocytes and sperm (Figure 4E). After fertilization, these gICRs generally remained at an intermediate methylation level in zygotic genomes (Figure 4E). When looking at the methylation of individual gICRs, we found that overexpression of Tet1 or Tet2 significantly reduced the methylation of 5 and 14 gICRs, respectively (Figure 4F and Appendix A). Of those, 2 ICRs (Peg10/Sgce, Snurf/Snrpn) were specifically regulated by Tet1, 10 were specifically acted on by Tet2, and 3 (Zrsr1/Commd1, Grb10, and Impact) were regulated by both Tet1 and Tet2 (Figure 4E,F). Tet1 also increased the methylation of Airn/Igf2r and Mest_(Peg1).

Careful examination of four representative gICRs, of which two were maternally imprinting (Snurf/Snrpn and Kcnq1ot1) and two were paternally imprinted (H19 and Rasgrf1), suggested that the GFP group (only Tet3 dominant) could sustain most methylated sites inherited from the parents. However, there was still partial removal of the methylation at some sites (Figure 4G,J). We found that Tet1 and Tet2 might perform erasure of imprinting at original imprinted sites, such as Rasgrf1 and Snurf/Snrpn (Figure 4H,I), but might maintain the methylation at some of the imprinted sites that were physiologically demethylated in the control (GFP group), such as H19 and Kcnq1ot1 (Figure 4G,J).

### 2.8. Genes with Altered Promoter Methylation by Tet1 and Tet2

It is well accepted that promoter methylation levels are associated with gene expression, and abnormal expression of genes in zygotes directly affects zygotic genome activation and embryonic development [32,33]. Here, we examined the genes with abnormal promoter methylation caused by Tet1 and Tet2.

In the Tet1 group, 536 genes with abnormal promoter methylation were identified, with 305 having hypomethylated promoters and 231 having hypermethylated promoters (Appendix A). Gene Ontology (GO) analysis suggested that the genes with hypomethylated promotors were mainly involved in the biological process (BP) for mRNA surveillance, embryonic morphogenesis, mitochondrial gene expression, and activation of GTPase activity (Figure 5A). The genes with hypermethylated promotors were mainly involved in the BP of cristae formation and cellular components (CC) of the MICOS complex (Figure 5A).

In the Tet2 group, the promoter methylation levels of the majority of genes were significantly downregulated, possibly facilitating the activation of some certain genes. A total of 2474 genes with abnormal promoter methylation were detected, of which only 23 genes (Appendix A) were considerably hyperregulated with no significant functional enrichment, while 2451 genes were identified as promoter hypomethylated (Appendix A) and were enriched in the BP concerning cellular response to various stimuli, including phenylalanine, lipid, protein kinase B, and so on (Figure 5B).

### 2.9. Early Expression of Tet1 or Tet2 Damages Embryonic Development

Next, we explored the effects of Tet1 and Tet2 overexpression on the developmental competence of zygotes (Figure 6A). Compared to the GFP-injected embryos, Tet2-injected embryos showed significant developmental impairment early at the four-cell stage, whereas Tet1-injected embryos showed a reduced developmental rate at the morula stage (Figure 6B). Both Tet1 and Tet2 injection led to lower blastocyst development than GFP injection control (Figure 6B). Even though a small proportion of embryos developed into blastocysts in the Tet1 and Tet2 groups, the blastocysts were smaller, with fewer cell numbers than embryos in the GFP group (Figure 6C,D). These results demonstrated that overexpression of Tet1 and Tet2 in zygotes was harmful to embryo development.

## 3. Discussion

During early embryogenesis, Tet3 is the sole Tet protein that exerts a demethylation function in zygotes, while Tet1 and Tet2 are expressed at later embryonic stages [14,16,19]. In this study, we investigated the influence of Tet1 and Tet2 on parental methylation in zygotes by ectopically expressing Tet1 and Tet2 in pronucleus-stage embryos. Our data suggested distinct effects of Tet1 and Tet2 on the parental genomes.

All three Tet family members have a similar ability to catalyze 5mC to 5hmC and to induce DNA demethylation [6,7]. Our immunofluorescence staining revealed that unlike Tet3, which has a clear preference for paternal DNA, ectopically expressed Tet1 and Tet2 were able to act on maternal DNA, promoting the 5hmC generation in the maternal pronucleus. Unexpectedly, however, we observed no gross 5mC change in the Tet1- or Tet2-overexpressed zygotes when detected via immunostaining.

To further explore the action of Tet1 and Tet2 on zygotic demethylation, we examined single-base-resolved methylation using WGBS. The analysis showed that both Tet1 and Tet2 could enhance the demethylation of the paternal genome and that the demethylation occurred across the genome, including genic, intergenic, and repetitive sequences, without an evident distribution bias among genetic elements. Tet3 was also shown to have a demethylation effect on various genic elements and repeats in the paternal genome, but it had minimal impact on LTR retrotransposons, such as ERV1 and IAP [12]. Our data suggested that Tet1 and Tet2 could reduce the methylation of these transposons in the paternal genome, indicating more extensive demethylation targets of Tet1 or Tet2 than Tet3. Overall, the exogenously expressed Tet1 or Tet2 could induce the demethylation of many genomic sites that are not targeted by endogenously expressed Tet3 in zygotes.

Although Tet1 could efficiently demethylate the paternal genome, Tet2 exerted more potent demethylation activity and yielded more demethylated sites than Tet1. However, only a small proportion of hypomethylated sites (on average 35.26% and 25.24% for genic regions and repeats, respectively) caused by Tet1 overlapped with those caused by Tet2, which suggests that Tet1 and Tet2 had different targets in most cases. In particular, Tet2 could also demethylate the maternal genome. The reason for this differential demethylation activity between Tet1 and Tet2 is unclear. Only the catalytic domain of Tet1 was used in our study, and the protein lacked the N-terminus containing the DNA-binding CXXC domain, which may reduce the chromatin binding capacity of Tet1 and impair its oxidation effect [34].

However, the exact role of CXXC in Tet proteins is still not fully understood. The catalytic domain alone can efficiently enter the nucleus, and create robust DNA oxidation [6,7]. A short isoform of Tet1, which lacks the CXXC domain, can also bind to the CpG islands across the genome, but with reduced targeted binding compared to the full-length Tet1 isoform that contains the CXXC domain [34]. In the case of Tet2, the single Tet2 isoform lacks the CXXC domain, but partners with a separating CXXC protein, IDAX, which interacts with Tet2 and recruits Tet2 to its genomic targets [35]. Tet3 also has two short isoforms without the CXXC domain [36]. The expression patterns, enzymic activity, and genomic binding specificity appear to be quite different among these Tet proteins and their isoforms. Full-length Tet1 is expressed in mouse embryos and PGCs [34], but only a short Tet3 isoform is expressed in the mouse oocytes [37]. Although lacking the CXXC domain, the oocyte-specific Tet3 has stronger demethylation activity than full-length Tet3 [37]. Thus, the existence of the CXXC domain appears not to be required for Tet proteins to induce genome-wide demethylation in zygotes. It is likely that the genomic distribution of Tet proteins may be aided by other factors that exist in oocytes. However, different Tet proteins may exhibit different oxidation potency, regardless of the absence of the CXXC domain. We found that even in the same zygotic environment, the catalytic domain of Tet2 had a stronger demethylation activity than that of Tet1. This suggests that other elements or the structure of Tets may have an influence on the function of the proteins. Indeed, the activities of Tet proteins are regulated by several mechanisms at different levels [38].

Although Tet2 could act on both paternal and maternal genomes to induce global demethylation, the reduction in methylation on the maternal genome was less than that on the paternal genome. This asymmetric action of Tet2 on the parental genomes was reminiscent of Tet3′s function. Tet3 also has a demethylation effect on the maternal genome, but to a much lesser extent than on the paternal genome [12,17]. This might reflect some inherent differences between the maternal and paternal genomes, and the maternal genome is more refractory to demethylation than the paternal genome. Given that Stella has a protective effect on genomic demethylation in the maternal pronucleus [18,39], it might also function in Tet2-induced demethylation. A previous study has shown that Stella interacts with Tet2 and Tet3, but not Tet1 [40]. However, the possibility of Stella’s involvement in this event needs further verification. A recent study reveals that Stella plays a critical role in preventing DNMT1-mediated de novo methylation, but not demethylation, during oocyte development [41]. In addition, depriving of H3K9me2 in oocyte chromatin has minimal effects on the maternal DNA methylation in mouse zygotes [42]. Thus, Stella and H3K9me2 may not contribute to the differential demethylation between the parental genomes. Other factors may have an influence on the action of Tets in pronuclei; for instance, the paternal chromatin is usually more decondensed than the maternal, and it may be more accessible to regulators.

The most unexpected observation was that Tet1 caused a slight methylation increase in the maternal genome. The increased methylation was mostly enriched in intergenic regions, including some transposal sequences, and occurred to a lesser extent in genic elements. We noted that the maternal genome in the control zygotes exhibited a slight methylation increase during the transition from oocyte to zygote. Despite the global loss of methylation in the zygotic pronuclei, de novo methylation has been observed in a subset of regions in the paternal genome, and it is believed that DNMT3A and DNMT1 are required for such de novo methylation [43,44]. DNMT3A was located in both parental pronuclei [44], implying that de novo methylation might also occur in the maternal genome. In this study, we found new methylated sites arising in the maternal genome after fertilization. However, it is difficult to explain the increased maternal methylation in Tet1-overexpressing zygotes, as DNMT3A and Tet1 compete with each other in binding DNA and regulating promoter methylation, as observed in mouse ESCs [45]. In contrast with the situation in the maternal genome, Tet1 caused global demethylation in the paternal genome, again reflecting the differences between the maternal and paternal genomes. However, we did observe substantial 5hmC generation in the maternal pronucleus in the detection via immunostaining. It should be noted that bisulfite sequencing cannot discriminate between 5hmC and 5mC [27]. Therefore, we cannot preclude the possibility that some 5hmC-deposited sites were read as “methylated” before being further oxidized or diluted by replication. The underlying mechanism needs further investigation.

A recent article suggested that Tet1 and Tet2 prefer the flanking sequence context of the CpG sites, with the strongest effects at the −1 and +1 positions [46]. In our study, we found that both Tet1 and Tet2 overexpressed in the zygotes more likely preferred the −5 to +5 flanking sequences, which is inconsistent with previous experiments performed in vitro [46]. Although Tet1 and Tet2 seemed to have slightly different flanking targets, they both preferred to act in CG-rich sequences with palindromic characteristics. Previous studies using ChIP-seq analysis in ESCs showed that Tet1 preferred to bind to high-CG-sequence environments [47]. These phenomena might be because methylation occurs symmetrically at CG loci with palindromic features in most cases [1,34]. Our results support previous observations that the short Tet1 isoform without CXXC domain could still prefer to bind to the CG islands [34].

gDMRs are a class of typical epigenetic modifications conferred by gametes. They undergo dynamic reprogramming during preimplantation development [29,30,31]. We observed more pronounced methylation changes in sperm-specific DMRs than oocyte-specific DMRs during the transition from gametes to zygotes; the methylation of sperm-specific DMRs was reduced in the paternal genome, but it increased in the maternal genome following fertilization. These data demonstrated that the observed intermediate methylation of sperm-specific DMRs in zygotes arose from both paternal demethylation and maternal remethylation. In contrast with sperm-specific DMRs, oocyte-specific DMRs retained methylation status in zygotic genomes. Both Tet1 and Tet2 enhanced the paternal demethylation and maternal remethylation of sperm-specific DMRs in zygotes, but they seemed to have fewer influences on oocyte-specific DMRs, although Tet2 reduced methylation to a degree. These results support that the maternal allele is more resistant to demethylation than the paternal one.

gICRs are regions typically determined by parent-specific DMRs and regulate gene imprinting [39,48]. gICR methylation patterns are established during germline cell development, and Tet1 and Tet2 have been shown to exert critical roles in imprinting erasure in PGCs [22,23,24,25,26]. As observed in early embryos, global 5mC decrease and 5hmC accumulation also occur during PGC development [26]. Both Tet1 and Tet2 are highly expressed in PGCs, but appear not to be required for the genome-wide demethylation in these cells [22,23,49], and they regulate the demethylation of some ICRs in a locus-specific manner [22,23]. During early embryogenesis, gICRs escape from the Tet3-induced demethylation in early embryos, which was suggested to be due to the protection by Stella [18]. In this study, we showed that overexpression of Tet2 significantly reduced the methylation levels of many gICRs, and that Tet1 specifically reduced the methylation levels of Peg10/Sgce and Snurf/Snrpn. In line with our results, previous studies show that demethylation of Peg10 and Snurf/Snrpn in PGCs is probably induced by Tet1 [22,23,24,25,34]. The specific role of Tet2 in regulation of ICR demethylation remains elusive. Tet2 may have a compensatory effect on Tet1, but it probably has different ICR targets [23]. Our data support that Tet2 can function in demethylating the imprinted genes. However, the methylation of some ICRs, such as the classic H19 and Igf2r, was not significantly downregulated by either Tet1 or Tet2. This may reflect some cellular differences between zygotes and PGCs. Additionally, the CXXC domain may be required for the recruitment of Tet1 and Tet2 to the target locus.

There are some controversies regarding the importance of Tet-mediated DNA demethylation or DNA oxidation for embryo development. Knockout studies have suggested that Tet protein members exert distinct roles at different developmental stages. Single, double, or triple deletion of Tet members leads to different developmental defects at post-implantation, perinatal, or neonatal stages, which is thought to be correlated with various aberrant hypermethylation phenotypes [16,19,20,21,22,50,51]. In most of these knockout cases, Tet proteins appear to be dispensable for preimplantation development. Interestingly, a recent study showed that morpholinos-mediated knockdown of Tets resulted in severe developmental defects of preimplantation embryos [52]. The authors suggested that the non-catalytic effects of Tet proteins, especially for Tet2, have critical roles in early embryogenesis [52]. So far, the data on overexpression of Tet proteins in embryos are scarce. In this study, we indicated that overexpression of Tet1 and Tet2 in zygotes impaired preimplantation development, which is quite different from the developmental phenotypes observed in Tet-deletion embryos. The compromised embryo development is probably associated with dysregulation of genomic methylation induced by Tet1 or Tet2.

GO-term analysis of the genes with abnormal promotor methylation suggested that the genes related to functions, such as energy consumption and stress response, might have altered their expression and affected embryonic development. In addition to the promoters, hypomethylated or hypermethylated sites also resided in gene bodies and repetitive elements, such as transposons. These elements play important roles in gene expression, genomic stability, and gene imprinting, which are critical for embryo development [53,54,55]. Although the potential impact of abnormal genomic methylation needs to be further validated, our study demonstrates that induced hypomethylation in the parental genomes at the zygotic stage by overexpression of Tet1 or Tet2 is detrimental to early embryo development. This seems to be consistent with the observed phenotype of Stella deficiency, in which genome-wide demethylation of the maternal genome results in embryo arrest at early cleavage stages [39,56]. Since genomic hypermethylation in zygotes induced by Tet3 knockout does not influence preimplantation development [16,57,58], abnormal hypomethylation of zygotic genomes appears to have a more negative effect on embryonic development than hypermethylation, as suggested by the present study.

## 4. Materials and Methods

### 4.1. Mice

All mice were housed under a 12 h light:12 h dark cycle and provided with food and water ad libitum. For parental genome splitting analysis by SNPs, male DBA/2J and female C57BL/6J mice were used to produce hybrid embryos.

### 4.2. Embryo Collection and Culture

Female mice aged 4–6 weeks were superovulated by injecting 10 IU human chorionic gonadotrophin (hCG) 48 h after injection of 10 IU pregnant mare serum gonadotrophin (PMSG), and then mated with males immediately. Eighteen hours later, the fertilized zygotes were collected from oviducts. The zygotes were briefly cultured in potassium simplex optimization medium (KSOM) in humidified 5% CO_2_ at 37 °C.

### 4.3. In Vitro Transcription and Microinjection

The coding DNA sequences for the catalytic domain [59] of Tet1 and Tet2 were cloned from mouse ovary mRNAs. For in vitro transcription, coding sequences for GFP, Tet1-CD, and Tet2-CD were cloned into the T7-driven vector pMDTM18-T (TaKaRa, Kusatsu, Japan). The mRNAs of GFP, Tet1-CD, and Tet2-CD were synthesized in vitro using the T7 RiboMAX™ Express Large Scale RNA Production System (Promega, Corporation, Madison, WI, USA) and m7G Cap Analog (Promega) following the manufacturer’s instructions. The final concentration of mRNA was diluted to 100 ng/μL before injection. The mRNA was injected into the cytoplasm of zygotes using an Eppendorf-driven micromanipulator (Eppendorf AG, Hamburg, Germany) under an inverted fluorescence microscope (Nikon TE300, Tokyo, Japan), with ∼10 pL of mRNA injection for each zygote. After injection, the embryos were cultured in KSOM for 12 h and then collected for immunostaining or methylome sequencing. To evaluate development, the embryos were continuously cultured to blastocysts. For the samples for single-cell bisulfite sequencing (scBS-seq), the polar bodies and attached sperm were removed from the embryos using a microtransplant needle and then stained with 10 μg/mL DNA dye Hoechst 33342 to confirm the pronucleus state.

### 4.4. Immunofluorescence Staining

Embryo samples were fixed in 4% paraformaldehyde [60] for 1 h at room temperature and permeabilized with 0.5% Triton X-100 in phosphate-buffered saline (PBS) for 30 min. To analyze 5hmC and 5mC, the samples were exposed to 2 M HCl for 15 min at room temperature and then neutralized in 0.5 M Tris-HCl (pH 8.5) for 10 min. Samples were blocked with 1% BSA and 0.2% Triton X-100 in PBS for 2 h and then incubated with primary antibodies overnight at 4 °C, followed by incubation with secondary antibodies for 1 h at room temperature. The nuclei were stained with 4,6-diamidino-2-phenylindole dihydrochloride (DAPI). Images were captured using an Olympus BX51 epifluorescence microscope (Olympus Corp, Tokyo, Japan). The primary antibodies for immunostaining included rabbit anti-5hmC (Active Motif, Carlsbad, CA, USA), mouse anti-5mC (Eurogentec, Liege, Belgium), mouse anti-H3K9me2 (Abcam, Cambridge, UK), and rabbit anti-Dppa3 (Santa Cruz Biotechnology, Santa Cruz, CA, USA).

### 4.5. Single-Cell Library Preparation

Pooled samples of sperm (n~100), metaphase II (MII) oocytes (n = 26), and embryos at the pronucleus stage from the GFP group (n = 29), Tet1 group (n = 23) and Tet2 group (n = 21) were collected.

We followed the protocol of library preparation by the previous reference to prepare the single sperm/oocyte/embryo WGBS library [61]. Genomic DNA was extracted using a Genome Extraction Kit (ZYMO, Orange, CA, USA). Prior to scBS-Seq, 2× embryo lysis buffer (10 mM Tris-Cl, pH 7.4, 2% SDS) containing the corresponding amount of λDNA as an internal reference and 1.0 μL proteinase K were added (final volume 20 μL), followed by incubation at 55 °C for 120 min for thorough lysis. Bisulfite treatment to convert unmethylated cytosines to uracils was conducted using the EZ DNA Methylation-GoldTM Kit ZYMO. Chemical denaturation was followed by incubation at 98 °C for 10 min, 64 °C for 160 min, and 4 °C for 10 min. For first-strand synthesis, 50 U of Klenow Exo– (Sigma, St. Louis, MO, USA) and dNTPs (1 nmol) were added to the mixture of transformed DNA and biotinylated random primer oligo1, followed by five random extensions at 4 °C for 5 min and 37 °C for 30 min. Agencourt Ampure XP beads (0.8×, Beckman Coulter) were used for purification and were washed twice with Tris-Cl (pH 8.5). Another random primer oligo2 completed the synthesis of the second strand under the conditions of 4 °C for 5 min and 37 °C for 90 min. The PCR procedure for amplifying the libraries was as follows: 95 °C for 2 min, 12–13 repeats of 94 °C for 80 s, 65 °C for 30 s, 72 °C for 30 s, 72 °C for 3 min and 4 °C, and then sc-BS library construction was completed. Final libraries were sequenced on an Illumina HiSeq 2500 in rapid-run mode with 100 bp paired-end reads.

### 4.6. Data Processing and Analysis

All raw bisulfite sequencing reads were trimmed to remove adaptors and low-quality bases using Trimmomatic (version 0.39; parameters: -clip_r1 9 -clip_r2 9 -paired HEADCROP:9 TruSeq3-PE-2. fa: 2:30:10 SLIDINGWINDOW: 4:15 LEADING:3 TRAILING:3 MINLEN:36). The trimmed sequences (filtered reads) were first mapped to the mouse genome (Mus musculus. GRCm39) using Bismark21 Methylation Extractor (version 0.22.3; parameters: -pe, -bowtie2, -non-directional, -unmapped). The Deduplicate Bismark tool was used to remove duplicate reads.

The Bismark21 software was used to extract individual CG loci of each sample. CG methylation levels were considered to be the average methylation at each CpG position. Single-CpG methylation levels were limited to CpGs with at least fivefold coverage. For the data after splitting using SNPs, reads for all the CpGs that were covered more than threefold were used to estimate the methylation level as described for single CpGs. Transposon element (TE) and repeat sequence annotation required the use of Repeatmask software (version: 4.1.2; parameters: -species “mouse” -pa 20 -poly). Gene functional regions were annotated by genomic “gff” files, and the longest transcripts were chosen. The Bioconductor package DSS was used to identify DMRs with *p*-value ≤ 0.01, CpG sites ≥ 3, DMR length ≥ 50 bp, and a smoothing window of 100 bp. Differentially methylated CpG dinucleotide positions (CG positions) were determined using the DSS package (FDR < 0.05), and 10 bp bases before and after this position were extracted for flanking sequence analysis. The frequency of bases presented at each position was counted and described as observed/expected (obs/exp). The variance was used to assess the propensity of Tet protein acted sites. Visualization and analysis of the data were performed using R scripts.

Gene Ontology (GO) functional enrichment analysis was performed for genes where the DMR overlapped with the promoter on the annotation results of the DMR on the genome. The GO analysis was carried out using the Metascape online tool (https://metascape.org/gp/index.html#/main/step1, accessed on 15 October 2021) [62]. GO terms with an FDR value (adjusted *p*-value) of less than 0.05 were considered to be significantly enriched.

### 4.7. Single-Nucleotide Polymorphisms (SNPs) Calling

C57BL/6J and DBA/2J strain mouse genomic data were obtained from the Ensemble data browser (https://asia.ensembl.org/index.html, accessed on 20 April 2021) and randomly interrupted into 150 bp segments by software. These segments were utilized in the software Bwa Mem (0.7.17) after comparison. SNP calling was performed by marking duplex reads using GATK software (version: 4.1.4.1) and the default parameters of the Haplotype Caller tool [63]. SNPs were screened in all sequencing expedition loci (number of reads ≥ 2), and the detected SNPs were filtered to remove SNPs with inconsistent SNPs at the same position that could not be tracked by sc-BS (C/T or A/G). Based on the interparental SNPs, parental reads were strictly split for the three groups of zygotes to distinguish between the parents. Briefly, reads detected in the same set of zygotes containing only alleles from the DBA/2J mouse marker were considered to belong to the paternal genome, and those from the C57BL/6J mouse marker were considered to be the maternal genome. The reads were discarded if both DBA/2J and C57BL/6J alleles were detected with the same reads. The final splitting efficiency ranged from 10% to 15%.

### 4.8. Statistical Analyses

GraphPadPrism5.0 software was used to make charts and pictures. R-language was used for Student’s *t*-test, Fisher’s test, and one-way analysis of variance. IGV software was used to realize the visualization of locus methylation levels.

## 5. Conclusions

In conclusion, our study demonstrates that artificially induced early expression of Tet1 and Tet2 in zygotes could substantially alter the zygotic methylation landscape and affect embryonic development. Tet1 and Tet2 yielded different consequences when acting on the parental genomes, which was mainly manifested by the greater demethylation potency of Tet2 than Tet1. These findings provide new insights into understanding the function of Tet dioxygenases and the mechanism of DNA methylation in relation to embryogenesis.

## Figures and Tables

**Figure 1 ijms-23-08495-f001:**
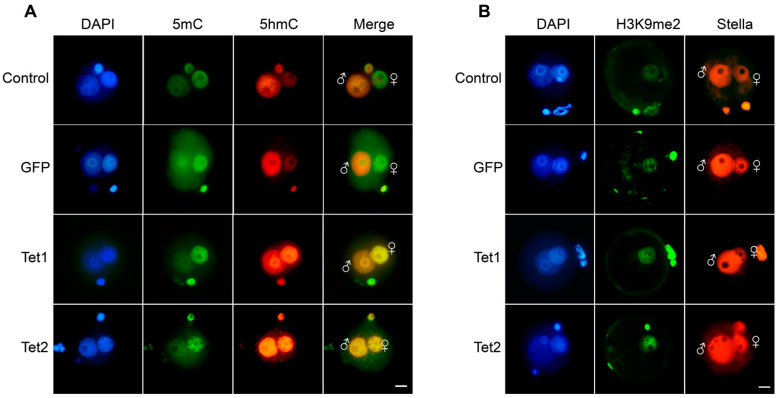
Methylation patterns of mouse zygotes microinjected with Tet mRNA. (**A**) Immunofluorescence staining for 5mC and 5hmC in zygotes injected with GFP (n = 39), Tet1 (n = 29), or Tet2 (n = 25) mRNAs. The non-injected zygotes were used as the control (n = 27). (**B**) Immunofluorescence staining for H3K9me2 and Stella in zygotes. “♀”, female pronucleus, “♂”, male pronucleus. Scale bar = 20 μm.

**Figure 2 ijms-23-08495-f002:**
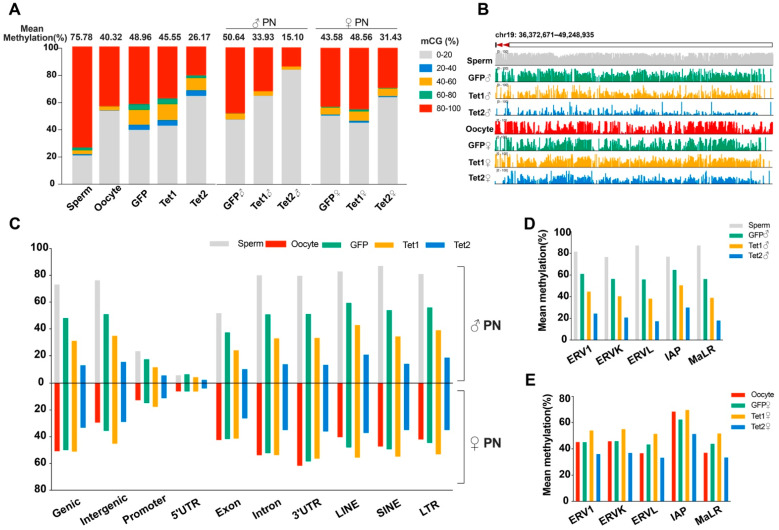
Methylation levels of parental genomes in gametes and zygotes. (**A**) A global view of DNA methylation in each sample. The proportions of 100 bp tiles at five methylation levels are shown. (**B**) Changes in the methylation levels of representative regions obtained using the IGV browser. (**C**) Mean DNA methylation levels of various functional regions and transposon regions. (**D**,**E**) The paternal and maternal genome methylation levels in subcategories of LTR repeats, respectively.

**Figure 3 ijms-23-08495-f003:**
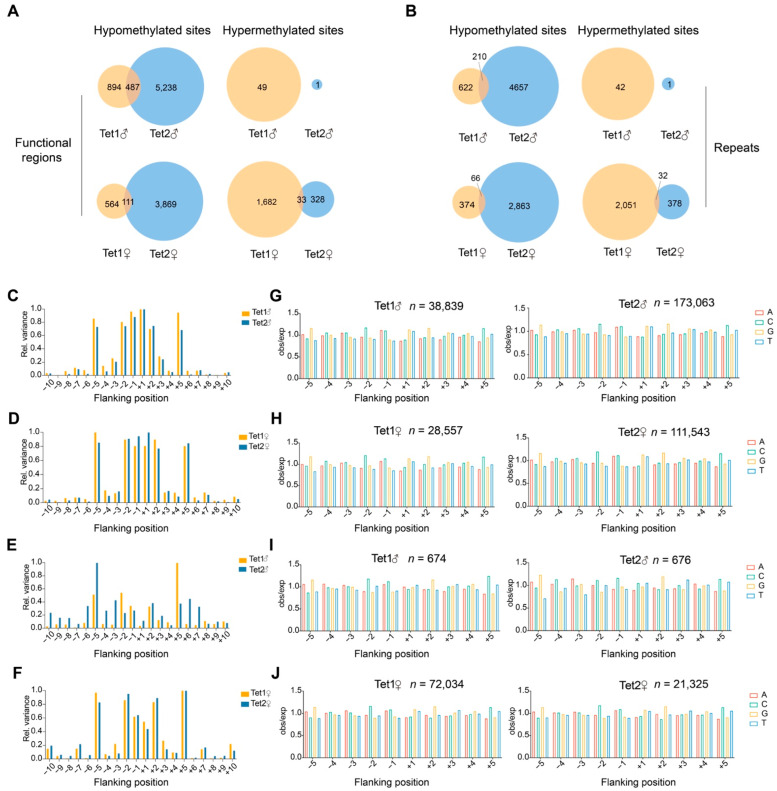
Analysis of Tet1- and Tet2-acted sites. (**A**,**B**) The numbers of methylation changed-sites from Tet1 and Tet2 injection compared with GFP injection. The DMRs overlapping region ≥ 1 bp and q-value < 0.01 were considered methylation overlapped sites. The term “hypomethylated sites” was defined by more than 20% methylation reduction than the GFP injection control, while “hypermethylated sites” were the opposite. (**C**–**J**) The propensity of Tet1 and Tet2 to act on sequences flanking CG. The observed/expected (obs/exp) values are presented as the frequency of occurrence of individual bases. The sum of the (obs/exp − 1)2 values of A, C, G, and T at CG positions context (−10 to +10), considered relative variation, were examined separately on the parental genome. The analysis of hypomethylated sites (**C**,**D**) and hypermethylated sites (**E**,**F**) is shown (hypomethylation and hypermethylation here represent a single CG position situation, FDR < 0.05). (**G**–**J**) Detailed base frequency for different positions of CG. “n” indicates the numbers of sequences analyzed.

**Figure 4 ijms-23-08495-f004:**
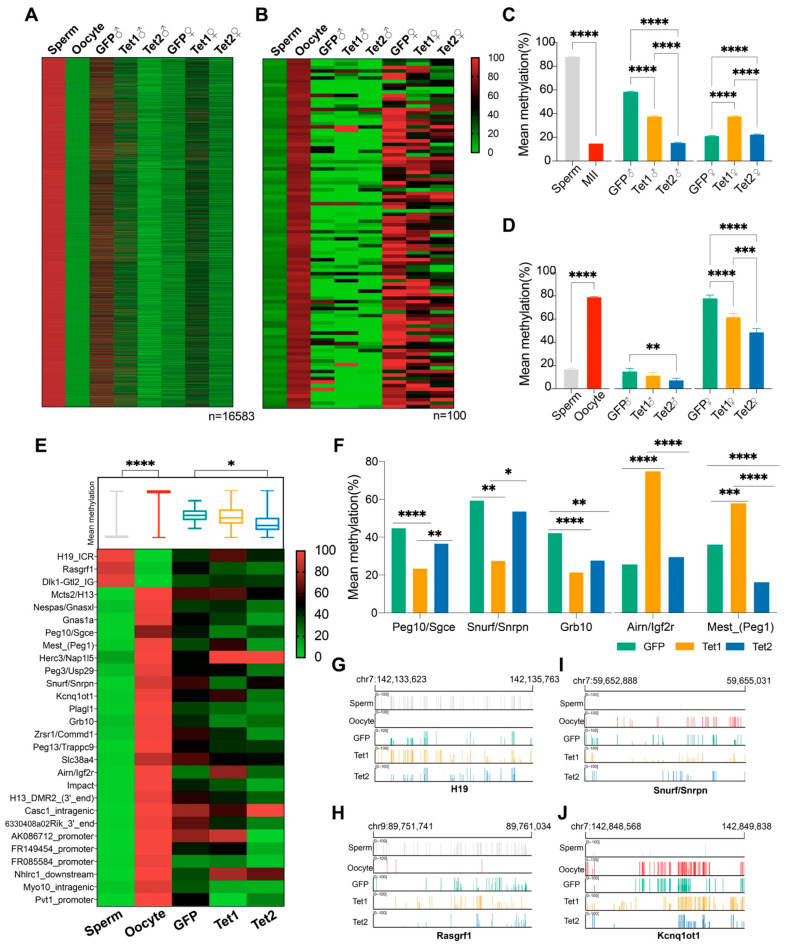
Effects of Tet1 and Tet2 on germline-specific DMRs. Germline-specific DMRs were defined as DMRs with ≥75% methylation levels in one parent and ≤ 25% in the other, and were examined for their presentation on the parental genome of zygotes. (**A**,**B**) Heatmaps for the methylation distribution of sperm-specific DMRs (n = 16,583) (**A**) and oocyte-specific DMRs (n = 100) (**B**). (**C**,**D**) Bar graphs for the mean methylation (±SEM) of the sperm-specific DMRs (**C**) and oocyte-specific DMRs (**D**) were examined in all samples. (**E**) Heatmap for the methylation levels of 25 maternal and 3 paternal known gICRs. The mean methylation of these gICRs from various groups is shown in the box graph above the heatmap. (**F**) Bar graphs for the mean methylation levels of 5 representative gICRs. (**G**–**J**) Graphical representation of methylation in paternally imprinted genes (H19 and Rasgrf1) and maternal imprinted genes (Snurf/Snrpn and Kcnq1ot1) at a locus in the gametes and zygotes. Different colors of short lines highlight the tracked methylated CpGs. “*”, *p* < 0.05; “**”, *p* < 0.01; “***”, *p* < 0.001; “****”, *p* < 0.0001.

**Figure 5 ijms-23-08495-f005:**
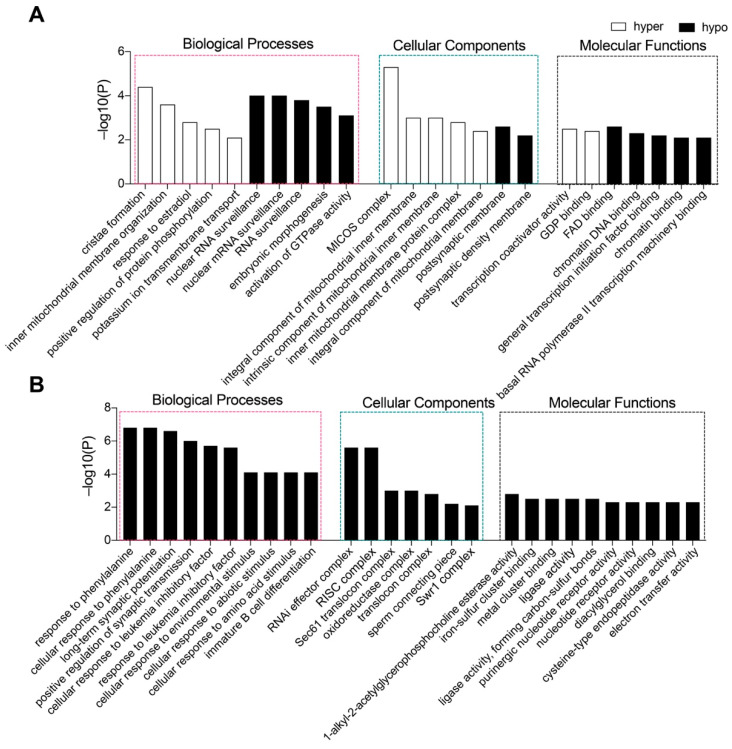
Genes with promoter methylation altered by the action of Tet1 and Tet2. (**A**,**B**) GO-term enrichment analysis of functions of genes with hypomethylated promoters caused by Tet1 (**A**) or Tet2 (**B**). The genes were enriched with criteria at −lg (*p*−value), *p*−value < 0.01.

**Figure 6 ijms-23-08495-f006:**
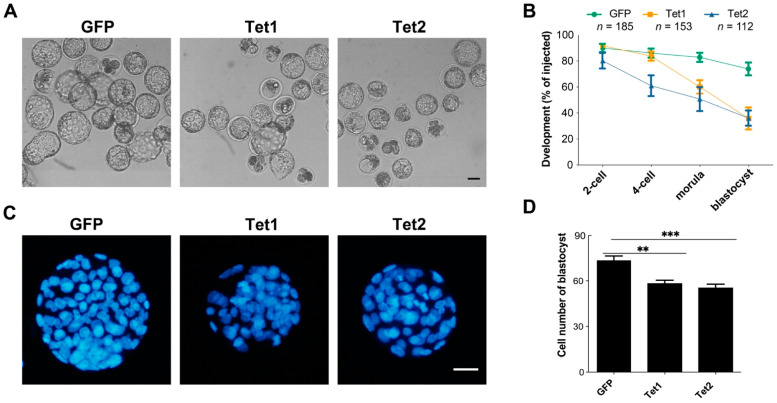
The effect of overexpression of Tet1 and Tet2 on embryonic development. (**A**) The blastocysts developed from zygotes injected with GFP, Tet1, or Tet2 mRNA. (**B**) Preimplantation development rates of the injected zygotes. (**C**) Representatives of blastocytes stained with DAPI for labeling the cells. (**D**) Statistical analysis of total cell number in blastocysts. The number of stained blastocysts in each group was more than 25. Scale bar = 100 μm. “**”, *p* < 0.001; “***”, *p* < 0.0001.

## Data Availability

All WGBS data that support the findings of this study have been deposited in the Sequence Read Archive (SRA) database under the accession code PRJNA830608.

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
