# Peer review of "Early Expression of Tet1 and Tet2 in Mouse Zygotes Altered DNA Methylation Status and Affected Embryonic Development"

_ijms, 2022, doi:10.3390/ijms23158495_

Round 1

Reviewer 1 Report

This is a very interesting story, which seems to improve the embryonic development knowledge of Tet1 and tet2 management. However, several empirical methods are needed:

1. In order to understand readers more easily, a summary diagram is needed.

2. The reference format must be consistent with the log policy (for example, the log name should be shortened to the end time).

3. Although experimental methods are allowed, the author must infect the main negative forms of Tet1 and tet2. Then, the author needs to ensure that these dominant forms do not have an important impact. This may be better than GFP alone.

4. The authors should confirm the expression and protein profile of Tet1 and tet2 in normal embryonic development, not in overexpressed embryos.

5. The author must also compare the emporic development process with tet1/tet2 recording conditions

Reviewer 2 Report

Although the findings that artificially induced early expression of Tet1 and Tet2 in zygotes substantially altered the zygotic methylation landscape and affected embryonic development are very interesting, numbers of points need clarifying and certain statements require further justification. These are given below.

<Points>

1.      The authors described, “The procedure of animal experiments was in accordance with the animal care policies of China Agricultural University and was approved by the Animal Ethics Committee at the University” without approval date (lines 568-570). Please show the approval date.

2.      In line 4, the authors described, “Qi Qi*, Qianqian Wang*”. In lines 9-11, only one E-mail address was described. In addition, the authors described, “Jian Hou1,*” but, there is no “1”. Please make it clear.

3.      In line 446, “CO2” should be changed to “CO2 (subscript)”.

4.      In line 458, “Nikon TE300, Japan” should be changed to “Nikon TE300, Tokyo, Japan”.

5.      All the reference style should be matched in the Int. J. Mol. Sci. style. For example, “Smith, Z. D.; Meissner, A., DNA methylation: roles in mammalian development. Nat Rev Genet 2013, 14 (3), 204-20.” Should be changed to “Smith, Z.D.; Meissner, A. DNA methylation: roles in mammalian development. Nat. Rev. Genet. 2013, 14 (3), 204-220.”.

Author Response

Point 1.  The authors described, “The procedure of animal experiments was in accordance with the animal care policies of China Agricultural University and was approved by the Animal Ethics Committee at the University” without approval date (lines 568-570). Please show the approval date.

Response1 : Thanks to the reviewer. We have added the approval date (data of approval: September 18, 2017) in the revised manuscript.

Point 2. In line 4, the authors described, “Qi Qi*, Qianqian Wang*”. In lines 9-11, only one E-mail address was described. In addition, the authors described, “Jian Hou1,*” but, there is no “1”. Please make it clear.

Response2: Thank the reviewer very much for pointing out these mistakes. Qi Qi and Qianqian Wang are co-first authors and Jian Hou is the sole corresponding author. We have made corrections in the revised manuscript.

Point 3.  In line 446, “CO2” should be changed to “CO2 (subscript)”.

Response 3: Thank the reviewer very much. We have corrected it.

Point 4.  In line 458, “Nikon TE300, Japan” should be changed to “Nikon TE300, Tokyo, Japan”.

Response 4: Thanks to the reviewer. We have made a revision.

Point 5.  All the reference style should be matched in the Int. J. Mol. Sci. style. For example, “Smith, Z. D.; Meissner, A., DNA methylation: roles in mammalian development. Nat Rev Genet 2013, 14 (3), 204-20.” Should be changed to “Smith, Z.D.; Meissner, A. DNA methylation: roles in mammalian development. Nat. Rev. Genet. 2013, 14 (3), 204-220.”.

Response 5: Thank the reviewer very much. We have made revisions on the reference format following the format style of Int. J. Mol. Sci. .

Round 2

Reviewer 1 Report

Authors have fully addressed all issues. So, it is now acceptable.